# MODELBENCH: A BENCHMARK FOR EXTRACTING EXECUTABLE, PHYSICS-BASED MODELS FROM SCIENTIFIC LITERATURE

## ABSTRACT

We introduce **ModelBench**, a benchmark for evaluating whether AI systems can extract executable physics-based models from scientific literature. ModelBench couples (i) gold-standard reference models, (ii) a hierarchical, weighted binary rubric covering physics correctness, code quality, and reproduction quality, and (iii) a judge protocol that produces pass/fail scores at rubric leaves. Unlike code-generation benchmarks that test function-level correctness, ModelBench targets the end-to-end task of reconstructing physically grounded models from incomplete and underspecified scientific descriptions. We release the benchmark specification, rubric generator and judge prompts, and an initial set of 20 gold models within the field of photonic integrated circuits, alongside scripts for fully reproducible evaluation. Candidate systems are required to produce a Python implementation of the model, a plot of the fitted results, and evaluate MSE and $R^2$ metric of the fit. Using general-purpose LLMs as neutral baselines, we report aggregate scores and case studies that reveal common failure modes (e.g., constraint violations, phenomenological overfitting) and show how rubric structure aids in diagnostic evaluation. We discuss limitations (judge variance, dataset breadth, implicit-knowledge gaps) and outline a roadmap to expand domains, tighten constraint checking, and support multiple valid solutions. ModelBench provides a transparent platform for tracking scientific modeling capabilities in AI under physical and empirical constraints.

## 1 INTRODUCTION

Consider a physicist encountering a promising paper on a compact experimental system, such as an optical resonator, a thin-film solar cell, or a superconducting qubit. While plots and apparatus descriptions are often provided, critical implementation details are frequently omitted: calibration constants, loss mechanisms, and material properties at the relevant operation point that are known tacitly within the field (Bogaerts & Chrostowski, 2018). Reconstructing a working model thus becomes an exercise in inference: recovering equations from prose, identifying constraints like energy conservation or passivity, and reconciling units and scaling conventions. These challenges contribute to the broader reproducibility crisis in science (Baker, 2016; Ioannidis, 2005).

Recent large language models can generate executable code and, when combined into agentic systems, can chain tools to search, fit, and validate models (Chen et al., 2021; Austin et al., 2021; Yao et al., 2023; Schick et al., 2023). But the core task here is not code generation per se, it is model reconstruction: identifying a suitable physics-based model, encoding assumptions as equations, fitting interpretable parameters under domain constraints, and ensuring the result is auditable and reproducible. Despite progress in physics-informed learning and LLM-as-judge evaluation, the community lacks a widely adopted, end-to-end benchmark that measures how well different systems perform this literature-to-model workflow under consistent, reproducible criteria (Raissi et al., 2019; Karniadakis et al., 2021; Zheng et al., 2023; Zhuge et al., 2024).

We introduce **ModelBench**, a benchmark that evaluates whether AI systems can read a scientific paper and produce an executable physics model. ModelBench provides expert-built gold models, which serve as short reference implementations, and a scoring rubric organized into a set of binary

checks. The rubric tests physics correctness, completeness of the code and outputs, and the ability to reproduce key results. A judging script evaluates each rubric item, records a yes/no decision, and combines the results into a weighted score. This way, the benchmark enables systematic and comparable evaluation across methods by fixing inputs, scoring criteria, and reporting scripts.

We release the full benchmark in a public repository (GitHub): The initial tasks with gold models, all the LLM prompts for the model creation, judging and rubric creation, and baseline evaluation scripts to support transparent, community-driven iteration. We provide baseline runs to illustrate how to use the benchmark; our findings are reported in Section 5.

Table 1: ModelBench task at a glance

| Action | What the system must do |
| --- | --- |
| Read | Parse paper context and figure captions to understand the system. |
| Model | Implement a suitable physics model as executable Python code (`solution.py`). |
| Fit | Fit to experimental or digitized-from-figures data under physical constraints. |
| Report | Save fitted parameters and metrics (MSE, $R^2$) in `model_fit.json`, generate a plot of the results (`model_fit.png`). |

Our first release focuses on photonics, a subfield of physics that studies the behavior of light in devices and materials. We chose photonics because many systems are compact and well captured by standard equations, which makes constraint checks and parameter interpretability easier to test. The benchmark design is domain-general and can be applied to other areas of physics and engineering.

The benchmark is intentionally tool-agnostic. AI systems receive the same paper context, experimental data, and a modeling query, and they must return an executable function along with fitted parameters that satisfy physical constraints. Our default setting is non-interactive to emphasize autonomous modeling ability (no human in the loop). The goal is to measure methodological competence in scientific modeling: choosing an appropriate model class, enforcing constraints, recovering interpretable parameters, rather than solely optimizing an expressive blackbox model that lacks physical interpretability. In this way, ModelBench provides a foundation for evaluating scientific AI in realistic, underspecified settings. We will continue to expand the dataset, extend to new domains, and invite community contributions.

## 2 RELATED WORK

Recent benchmarks increasingly probe the limits of LLMs in code generation and research reproduction. PaperBench introduced a hierarchical rubric framework for grading AI replications of machine learning research code, using LLMs as judges (Starace et al., 2025). General-purpose code generation tasks such as HumanEval (Chen et al., 2021) and SWE-bench (Jimenez et al., 2024) instead target narrow programming skills-isolated functions or bug fixes where success is defined by functional correctness in a restricted scope. In contrast, ModelBench addresses a full end-to-end modeling workflow: extracting knowledge from complex scientific literature and producing executable, physics-based models. This setting demands holistic domain reasoning that extends well beyond function-level correctness.

Our evaluation approach also builds on emerging LLM-as-a-judge methodologies. MT-Bench showed that LLMs can reliably score and compare responses when guided by consistent rubrics (Zheng et al., 2023), and the Agent-as-a-Judge framework extended this idea to rubric-based multi-agent assessments for complex tasks (Zhuge et al., 2024). In parallel, physics-informed machine learning motivates our domain-specific emphasis. Approaches such as physics-informed neural networks (PINNs) embed physical laws directly into neural architectures (Raissi et al., 2019), while surveys of scientific machine learning highlight the importance of enforcing real-world constraints during training (Karniadakis et al., 2021). These methods typically assume well-specified governing equations or clean datasets. ModelBench instead targets the harder problem of inferring and executing models from incomplete textual and tabular descriptions, thereby bridging the gap between function-level code-generation benchmarks and physics-centric learning.

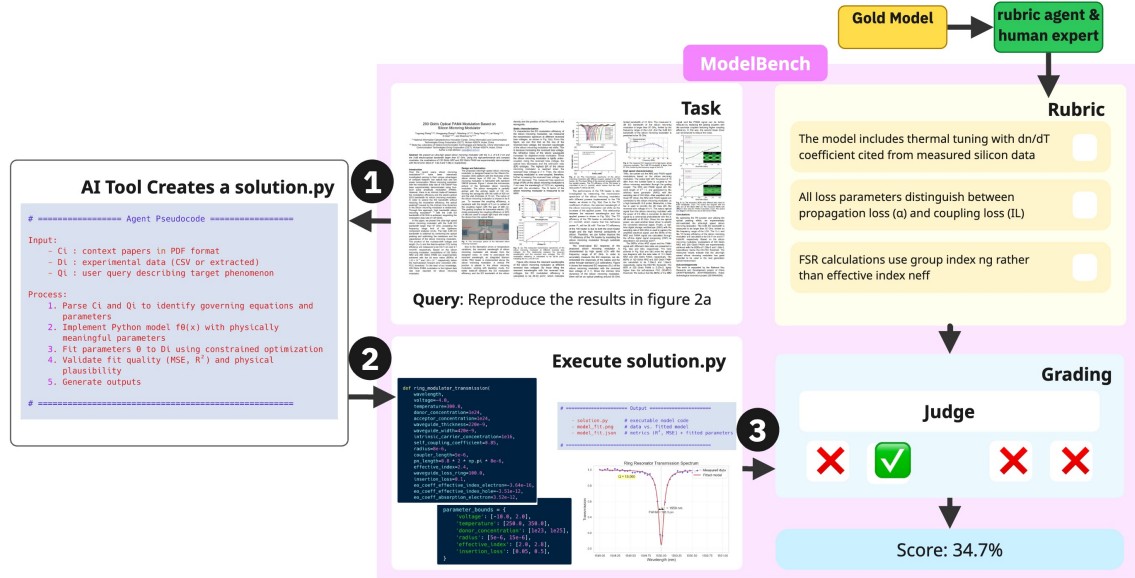

Figure 1: Overview of the ModelBench evaluation pipeline. Given a paper context, experimental data, and a task query (1), an AI system generates a Python script (`solution.py`) that implements a physics-based model and produces outputs including fitted parameters (`model_fit.json`) and a comparison plot (`model_fit.png`) (2). These outputs are then checked against task-specific rubrics derived from expert-built gold models (3). A judge assigns pass/fail scores to each rubric item, which are aggregated into a final task score.

## 3 THE MODELBENCH FRAMEWORK

ModelBench provides a structured evaluation framework for assessing the scientific modeling capabilities of AI systems. It is composed of four components: (1) a precise modeling task with defined inputs and expected outputs; (2) a gold standard reference implementation authored by experts that captures the target physical behavior; (3) a hierarchical weighted rubric that defines the correctness and completeness criteria; and (4) a scoring protocol that maps rubric decisions to quantitative scores.

### 3.1 TASK DEFINITION: FROM PAPER TO EXECUTABLE MODEL

Each benchmark instance $i$ defines a computational modeling task grounded in scientific literature. The inputs consist of a paper context $C_i$ (typically a PDF document), experimental data $D_i = \{(x_{ij}, y_{ij})\}_{j=1}^{n_i}$ extracted from tables or digitized figures, and a task query $Q_i$ specifying the reproduction objective.

Given these inputs, the system must generate three artifacts: a Python implementation `solution.py` encoding the physical model and parameter estimation procedure, a JSON specification `model_fit.json` containing optimized parameters $\theta_i^*$ with associated performance metrics (MSE and $R^2$), and a visualization `model_fit.png` comparing model predictions to experimental observations. The benchmark is tool-agnostic, and, in the default configuration, we require single-shot inference, prohibiting iterative refinement or human feedback.

### 3.2 GOLD MODELS AND RUBRIC CONSTRUCTION

Domain experts construct a gold model $\bar{f}_{\theta,i}$ for each task, providing a canonical reference implementation that captures the essential physics described in paper $C_i$. From these models, structured evaluation rubrics $R_i$ are derived:

$$R_i = \mathcal{G}_{\text{rubric}}(\bar{f}_{\theta,i}, C_i, Q_i), \tag{1}$$

where $\mathcal{G}_{\text{rubric}}$ denotes the rubric generation process. Each rubric $R_i$ encodes a weighted hierarchy of evaluation criteria in JSON format, enabling automated assessment without requiring direct comparison to the gold implementation. For instance, a ring resonator rubric might assign 40% weight to correct Lorentzian transmission form, 30% to extracted parameters (free spectral range within 2% of gold, quality factor within 10%), 20% to physical constraints (coupling coefficient $\kappa^2 \leq 1$, positive propagation loss), and 10% to implementation quality (wavelength grid resolution, numerical stability near resonance). This decomposition ensures candidate solutions are evaluated on both quantitative accuracy and physical validity.

### 3.3 CONSTRAINT-AWARE PARAMETER FITTING

The benchmark requires systems to identify parameters that simultaneously minimize prediction error and satisfy physical constraints. The optimization problem takes the form:

$$\theta_i^* = \arg\min_{\theta \in \Theta} \text{MSE}(D_i, f_{\theta,i}(x)), \tag{2}$$

where the MSE objective provides a differentiable metric amenable to gradient-based optimization across diverse physical domains. The set $\Theta$ represents the feasible parameter space defined by physical admissibility conditions, such as positivity of dissipation coefficients, energy conservation, causality requirements, or passivity conditions. While these constraints may originate as general inequalities $g(\theta) \leq 0$ or equalities $h(\theta) = 0$, effective implementations often eliminate them through model reparameterization, such as exponentiating parameters to ensure positivity or using normalized representations to enforce sum-to-one constraints. Systems must autonomously determine appropriate constraint handling strategies, as the benchmark framework performs no automatic constraint enforcement. As detailed above, constraint satisfaction constitutes a distinct evaluation criterion, assessed independently of prediction accuracy during scoring. These constraints, which are essential for ensuring the model is physically meaningful, are derived from well-established principles of optics and quantum electronics (Yariv & Yeh, 2007; Haus & Huang, 1991).

### 3.4 SCORING AND JUDGING PROTOCOL

Submissions are evaluated against task-specific rubrics $R_i = \{r_1, ..., r_m\}$, where each criterion $r$ carries weight $w_r$ with $\sum w_r = 1$. Figure 2 illustrates the hierarchical structure spanning three categories: **Physics Correctness**, which verifies the selection of the model class, the adherence to constraints, and unit handling; **Completeness**, which ensures artifact generation, executability, and I/O conformance; and **Reproduction Quality**, which assesses the fit quality with the reported results.

A judge, which can be an automated script or another LLM, assigns a binary scores $e(r) \in \{0, 1\}$ to each leaf criterion, yielding the weighted score:

$$C_{\text{rubric},i} = \sum_{r \in R_i} w_r e(r). \tag{3}$$

Judge reliability is assessed through cross-validation with secondary judge variants and human audits on randomly sampled tasks, providing uncertainty estimates for the evaluation protocol.

This granular scoring enables partial credit and detailed performance analysis across physical modeling dimensions. All evaluation infrastructure–gold models, rubric generators, judge implementations, and grading scripts–is publicly available to ensure reproducibility and facilitate community refinement (GitHub).

## 4 DATASET

The ModelBench dataset comprises 20 tasks from photonic integrated circuits, each centered on a paper reporting experimental characterization of compact devices–ring resonators, Mach-Zehnder modulators, or spectral filters. Domain experts implemented gold models capturing the essential physics through parameterized mathematical functions that reproduce reported experimental curves. These reference implementations anchor the evaluation rubrics and are publicly available in the GitHub repository (GitHub).

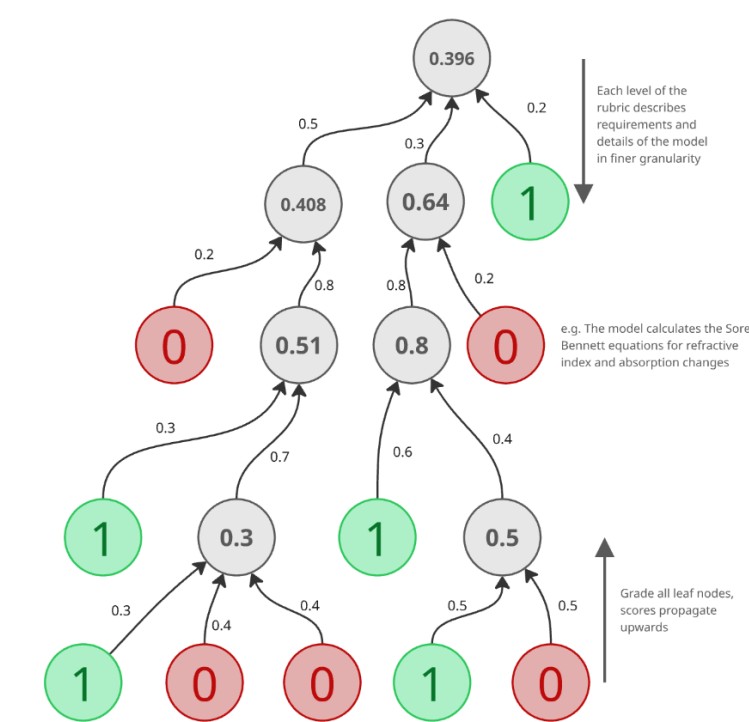

Figure 2: Hierarchical rubric structure showing categories and item weights.

**Example.** The following gold model implements wavelength-dependent through-port transmission through a ring resonator in the add-drop configuration, incorporating round-trip phase accumulation, waveguide loss, and coupling coefficients based on the S-matrix formalism:

```
def ring_resonator_transmission(wavelength_nm,
                                perimeter=1570.2e-6,
                                effective_index=2.4,
                                self_coupling_through=0.85,
                                self_coupling_drop=0.85,
                                waveguide_loss=233):
    wavelength = wavelength_nm * 1e-9  # Annotation [1]

    round_trip_length = perimeter  # Annotation [2]

    phi = 2 * np.pi * effective_index * round_trip_length / wavelength

    alpha_per_m = waveguide_loss * 100 / 4.343  # Convert dB/cm to 1/m
    a = np.exp(-alpha_per_m * round_trip_length / 2)  # Annotation [3]

    num_through = ((self_coupling_drop * a)**2 -
        2 * self_coupling_through * self_coupling_drop * a * np.cos(phi) +
        self_coupling_through**2)

    den = (1 -
        2 * self_coupling_through * self_coupling_drop * a * np.cos(phi) +
        (self_coupling_through * self_coupling_drop * a)**2)

    transmission_through = num_through / den  # Annotation [4]

    transmission_through = np.maximum(transmission_through, 1e-10)
```

```
transmission_through_dB = 10 * np.log10(transmission_through)
return transmission_through_dB
```

This example illustrates the typical level of abstraction: each gold model is only a few dozen lines of code, yet it encodes the central physical assumptions of the paper and provides a reproducible baseline for evaluation.

**Comparison to PaperBench.** PaperBench addresses underspecification by providing addenda with clarifications from the paper. In contrast, ModelBench does not add clarifications: agents must contend with the same underspecified and implicit assumptions that human physicists face when reconstructing a model from a paper through physically grounded reasoning. This design explicitly probes whether AI systems can bridge gaps in scientific reporting by choosing plausible parameterizations and enforcing physical constraints.

**Curation effort.** The implementation of the gold model by physicists constituted the primary effort to develop the data set, ensuring physical accuracy and appropriate abstraction levels. Rubric generation followed a two-phase protocol: Initial LLM-based formulation followed by expert refinement to guarantee the validity of the evaluation. The rubric generation prompts and complete curation pipeline are documented in the public repository. Future releases will expand coverage within photonics and extend to additional physics and engineering domains.

## 5 Experimental Setup and Results

As neutral baselines, we evaluated two general-purpose LLMs, OpenAI's GPT-5 and Anthropics Claude Opus 4.1, using identical prompts (all published on our Github (GitHub)) and environment. Both models successfully produced executable code and outputs for most tasks. Aggregate scores, averaged across the 20 tasks (each repeated 5 times per model for better statistics), are reported in Table 2.

Table 2: Baseline performance on ModelBench. Scores represent the fraction of rubric requirements satisfied, averaged over 20 photonics tasks. We show here the standard deviation as the error to highlight the spread of the scores.

| System | Score |
|---|---|
| ChatGPT-5 | $39.2 \pm 17.9\,\%$ |
| Claude Opus 4.1 | $27.9 \pm 13.0\,\%$ |

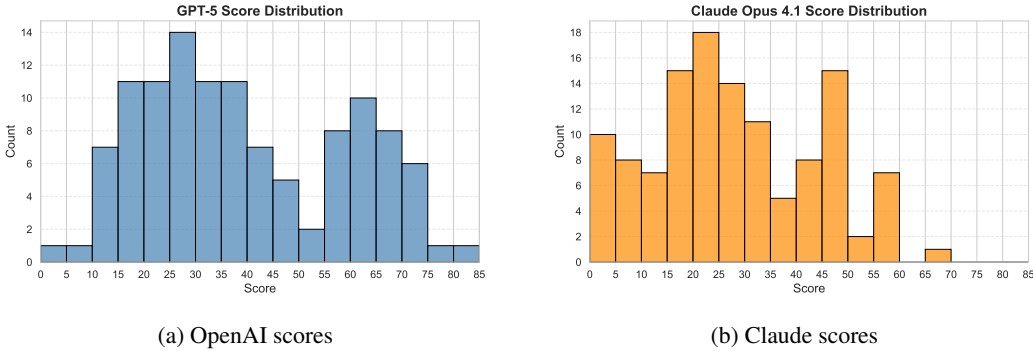

(a) OpenAI scores                              (b) Claude scores

Figure 3: Score distributions for OpenAI and Claude: We repeat the benchmark run 5 times for each model (GPT-5, Claude Opus 4.1) and plot the distribution of how high the models score.

# 6 DISCUSSION

## 6.1 BASELINE PERFORMANCE AND FAILURE MODES

Our baseline experiments with Claude Opus 4.1 and ChatGPT-5 provide a neutral reference point for the benchmark. Scores in the range of 40–60% indicate that general-purpose LLMs can often produce runnable code, but struggle to meet the full set of rubric criteria. Typical failure cases include: (i) *poor fitting performance*, where the model class is expressive enough but the fitting routine fails to recover correct parameters; (ii) *loss of scientific context*, where generated solutions ignore the physics described in the paper and instead adopt either oversimplified or unnecessarily elaborate models; and (iii) *incomplete outputs*, where required artifacts such as `model_fit.json` or plots are missing or incorrectly formatted. These failure modes highlight the gap between generic code generation and genuine scientific modeling competence.

Understanding these failure modes suggests several directions for improvement. First, agents need access to stronger fitting capabilities. Current baselines often rely on boilerplate open-source routines that do not reliably converge to physically meaningful solutions. Providing more robust and specialized fitting tools could improve parameter estimation. Second, baseline systems frequently lose track of the scientific context. Although they may achieve a reasonable numerical fit, the models themselves are either too simplistic (non-physical) or unnecessarily complex. Introducing an explicit planning step, where the system first outlines the physical principles relevant to the paper before coding, could help align generated models with the intended scientific context. Third, some failures stem from execution issues such as typos, incorrect file paths, or data-loading mistakes. While infrequent, these still cause rubric items to fail. An interactive execution environment with real-time feedback on code and file handling would mitigate such trivial issues and prevent otherwise valid solutions from being penalized.

## 6.2 JUDGE AGREEMENT AND RELIABILITY

Because rubric evaluation depends on a judge, consistency and reliability are critical. Our rubric leaves are binary (pass/fail), which reduces ambiguity and simplifies aggregation. To estimate reliability, we employ a secondary judge variant and small-scale human audits. Variance remains an open issue: not all failures are equally clear-cut. For instance, a model may be physically correct at multiple levels of complexity, yet only some formulations are appropriate for the specific regime represented in the data. A natural next step is larger-scale human validation to calibrate and benchmark judge performance.

## 6.3 DATASET BREADTH AND DOMAIN EXPANSION

The current release focuses on photonics, chosen for tractability and interpretability. The 20 tasks in photonic integrated circuits cover resonators, modulators, and filters. This domain provides advantages: compact systems, well-characterized equations, and interpretable constraints. However, it is also a limitation: the narrow scope does not test generalization beyond photonics. Future versions of ModelBench will therefore expand into broader areas of physics and engineering, including condensed matter, mechanics, and materials science.

## CONCLUDING REMARKS

ModelBench provides a first step toward systematic evaluation of AI systems on end-to-end physics modeling tasks. The benchmark highlights current limitations of general-purpose LLMs, while offering a transparent and extensible platform for tracking progress. By focusing on physical validity and reproducibility, ModelBench helps shift the focus of AI benchmarking toward the activities that matter in scientific practice.

## 7 ACKNOWLEGEMENTS

USE OF LARGE LANGUAGE MODELS (LLMS)

In accordance with ICLR 2026 policy, we disclose that large language models (LLMs) were used during the preparation of this paper. Specifically, LLMs were employed to aid in polishing the writing (improving clarity, grammar, and flow) and for retrieval and discovery of related work (identifying and summarizing relevant references). The conceptual development, experimental design, analysis, and conclusions were carried out entirely by the authors.

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
