# OpenReview forum: "ModelBench: A Benchmark for Extracting Executable, Physics-Based Models from Scientific Literature"
_ICLR.cc/2026/Conference — Submitted to ICLR 2026_

### Official Review · Reviewer_rfsS · 2025-10-31

**Soundness:** 3
**Presentation:** 4
**Contribution:** 3
**Rating:** 6
**Confidence:** 3

**Summary:**

The authors propose ModelBench, a benchmark for evaluating end-to-end AI-based extraction of physical models from physics literature. In this first version of ModelBench, the authors choose a subset of 20 papers in photonics, from which experts extracted gold models (Python scripts) reproducing the experiments in these papers. A candidate submission consists of a model implementation (compared to the reference one), plots of the results and goodness fit metrics. For each paper, the benchmark contains the gold model along with a set of evaluation criteria (weighted, hierarchical rubrics with binary responses) that stress-test a candidate submission's correctness.
ModelBench is model-agnostic and can work using LLM-as-a-judge evaluation as well as human feedback to produce an evaluation score. The authors plan on extending ModelBench with more models from various domains of physics.

**Strengths:**

- This paper's main contribution is the evaluation gap it is filling. As argued by the authors, expert-based model extraction from physics literature is slow and costly, and while the physics community has steadily been adopting AI-based extraction tools for this purpose, the lack of gold standard evaluation benchmarks makes it hard to quantify the reliability of these methods. ModelBench aims to fill this gap by proposing such a benchmark.
- ModelBench can be seen as a harder, physics-oriented version of PaperBench. ModelBench is completely end-to-end, requiring models to work from unannotated physics papers. In this, it mirrors the task of human researchers who need to infer real-world parameters from the often incomplete/implicit descriptions found in the literature.
- The well-structured and model-agnostic nature of the benchmark ensures that evaluation can be systematically performed across various models and systems.
- A considerable amount of work has been put into extracting gold models from articles and carefully designing relevant evaluation points for each of them, making the data of the benchmark itself a valuable resource.
- ModelBench does not claim to solve the reproducibility crisis, but is a pragmatic approach to leveraging the implicit assumptions used in research articles. Tools like ModelBench could eventually contribute to solving that root cause by validating that a future AI model is able to fully leverage those hidden assumptions and parameters.
- Among the evaluation criteria are explicit checks that the produced model follows physical constraints, such as energy conservation, and can reproduce experimental results. This is missing in previous works, such as PaperBench (notably due to the inherent differences in the targeted domains)
- The experimental results on GPT-5 and Claude Opus 4.1 provide some initial insights into the limitations of current LLMs for scientific modeling.

**Weaknesses:**

- In the initial release, 5 out of 21 papers are not available in open-access. While 75% of the benchmark remains freely available, this could constitute a significant hurdle that makes the benchmark harder to use, and makes its installation non-automatable, and more time-consuming for future releases. In addition, building a benchmark for reproducibility that relies on non-reproducible (inaccessible) sources is contradictory and runs counter to the increasingly strong movement towards open science. There are (valid) justifications for this choice, but it is hard to understand why this limitation is not mentioned or justified in the paper.
- While being model-agnostic is a strength in terms of flexibility, it also comes with several issues: The evaluation process requires a judge, which can either be an automated script or another LLM, to assign binary scores to rubric items. The authors correctly acknowledge that judge variance remains an open issue. A more thorough investigation of the reliability and biases of using an LLM as a judge is needed. The authors use human audits as a first good step, but a more detailed analysis of inter-judge agreement and potential calibration methods would strengthen the evaluation protocol.
- Creating gold-standard models and detailed rubrics by domain experts is very labor-intensive. The resulting benchmark is of high quality, but this also makes scaling the benchmark to a large number of tasks and domains challenging, which is not discussed in the paper. A discussion on potential strategies to scale this process in the future would be beneficial to the paper (partial automation? streamlining the process using a crowdsourced platform? etc.).
- The hierarchical rubric provides a structured evaluation, and different criteria seem to be used for different papers. However, as the benchmark scales up, there is a risk that future AI systems could be specifically optimized to perform well on the rubric's criteria without necessarily achieving a deeper scientific understanding, which may eventually pose issues. The authors could consider incorporating more open-ended evaluation metrics or human-in-the-loop assessments to mitigate this risk.
- The initial release of ModelBench is focused on photonic integrated circuits, with a dataset of 20 tasks. This limitation is acknowledged by the authors, who mention future expansion plans. However, the current narrow scope may limit the generalizability of the findings to other domains of physics. Future versions should prioritize a broader range of physics and engineering problems to demonstrate the benchmark's versatility.

**Questions:**

- Was a consistent team of experts involved in validating the rubrics? If not, was any cross-validation process involved to ensure that different experts would create similar rubrics for the same paper?
- As mentioned above, scaling ModelBench up seems like a major hurdle. Do the authors plan to address this in future releases, and how? In particular with respect to the rubric generation mentioned above.
- Why were closed papers chosen as part of the initial release? Was this done for realism reasons (as much of physics literature is locked behind paywalls)? Are those papers fundamentally more relevant?

---

### Official Review · Reviewer_wzC2 · 2025-10-31

**Soundness:** 2
**Presentation:** 2
**Contribution:** 1
**Rating:** 2
**Confidence:** 4

**Summary:**

This paper proposes ModelBench, a benchmark evaluating whether AI systems can extract executable, physics-based models from scientific papers. Each task provides a paper excerpt and experimental data; models generate runnable Python code implementing a physically meaningful model, fit parameters, and report metrics (MSE, R²). The benchmark includes 20 expert-curated photonics tasks with gold models, hierarchical weighted rubrics, and judging protocols. GPT-5 and Claude Opus 4.1 achieved 39% ± 18% and 28% ± 13% rubric satisfaction, respectively.

**Strengths:**

- This work explores the challenge of reconstructing physics-based models from literature beyond function-level code generation in prior literatures.
- Clear pipeline with gold models, rubrics, and reproducible scoring protocols.

**Weaknesses:**

- Only 20 photonic-circuit tasks with evaluation restricted to two models (GPT-5, Claude Opus 4.1). This scale is insufficient for drawing generalizable conclusions about LLM capabilities.
- The discussion of related work is limited, focusing primarily on PaperBench and ModelBench without sufficiently situating the benchmark in the broader context of scientific modeling, code generation, and physics-informed learning.
- Minimal detail on rubric validation, inter-rater reliability, or quality assurance. Judge variance and limited human calibration raise concerns about score reproducibility.
- The work feels closer to a technical report than a rigorous benchmark study. The limited scale and validation make it unsuitable for publication at a major venue.

**Questions:**

- What was the selection criteria for the 20 papers, and how much expert time was required per task?
- What is the inter-judge agreement rate for rubric scoring?
- The paper claims to move beyond code generation to “scientific modeling.” How is this distinction operationalized and measured?

---

### Official Review · Reviewer_BZeV · 2025-11-01

**Soundness:** 3
**Presentation:** 3
**Contribution:** 3
**Rating:** 6
**Confidence:** 2

**Summary:**

odelbench proposes an end-to-end benchmark to assess the capability of AI systems in
extracting executable physics-based models from scientific papers. Each task is provided with
paper context and data; systems must implement a physics model in Python, fit parameters
under physical constraints, and produce metrics MSE, R^2, and a comparison plot. Evaluation
leverages expert gold models to derive a hierarchical weighted binary rubric covering physics
correctness, completeness (artifact and executability), and reproduction quality (fit). Baselines
with general LLMs show that code can often run, while adherence to physics or constraints and
reproducibility often fail, which motivates this benchmark as a standardized way of tracking
scientific modeling capability.

**Strengths:**

On soundness, the problem is well-motivated; the framework (inputs, required artifacts, rubric, scoring) is
specified clearly. The rubric design and constraint-aware fitting are reasonable and grounded in
domain principles. Baseline results (with distributions and variability) support the central claims
about current LLM limitations. Threats to validity (judge variance, domain breadth) are
acknowledged with concrete mitigation/roadmap.

On presentation, writing is clear; the pipeline/rubric figures communicate the workflow; the ring-resonator
example makes the abstraction level concrete. Prior work is positioned well (PaperBench,
HumanEval/SWE-bench, PINNs, LLM-as-judge).

On contribution, the benchmark targets an important, under-served evaluation capability: literature-to-model
with physical constraints and reproducibility. Using gold models to auto-derive rubrics is a
useful, reproducible idea.

**Weaknesses:**

On soundness, baselines are limited in diversity and ablations (e.g., planning vs. no-planning, different optimizers).

On presentation, the paper could be improved by a tighter, tabular summary of rubric categories/weights across several tasks and a short “failure gallery” with side-by-side artifacts.

On contribution, significance is currently bottlenecked by domain breadth (20
photonics tasks) and limited baseline analysis, but the design is extensible and the contribution
is likely valuable to ICLR.

More specifically,

- Domain scope. The initial dataset is narrow (photonics, 20 tasks); generalization to
other physics/engineering areas is mentioned but I don’t see it in the writing.
- Judge reliability. LLM-as-judge is used as a yes/no framework; while variance is
acknowledged, it’s not clear how we can track the variance of an LLM’s output across
many runs.
- Baselines: Limited baseline diversity and missing ablations (e.g., planning step,
constraint reparameterizations, optimizer/backends, data digitization noise).
- Multiple-valid-solutions. Many physics problems admit non-unique but valid
solutions; there isn’t anything included for field aware answers.

**Questions:**

1. How will the rubric accept multiple valid methods without false negatives? Is there a
system to handle this case?
2. How consistent are your graders? Can you share how often different judges agree when
scoring (even a simple % would help)?
3. If there are multiple correct ways to model the same system, how do you avoid
penalizing a valid but different solution?
4. Does adding a short “planning step” (write the physics assumptions first, code second)
improve scores?

---

### Meta-Review · Area_Chair_a423 · 2026-01-05

**Summary:**

This paper introduces ModelBench, a benchmark for evaluating whether AI systems can extract executable, physics-based models from scientific literature. Reviewers generally agreed that the problem is well motivated and that the benchmark design is clearly specified, including the use of expert-curated gold models, hierarchical rubrics, and reproducible evaluation protocols. The paper is clearly written, and the released artifacts are useful.

However, there were significant concerns about whether the current version of the benchmark is sufficiently mature and broad to warrant acceptance at ICLR. In particular, reviewers questioned the limited scale of the benchmark, the narrow domain focus on photonic integrated circuits, and the lack of strong validation of rubric reliability and judge consistency. While the authors provided clarifications in the rebuttal, no reviewer replied after the rebuttal to indicate that their concerns had been resolved or that they intended to change their score. Based on the remaining concerns and the overall review signal, I recommended rejection.

**Reviewer Concerns:**

A key concern across reviews is that the benchmark is currently too limited in scope to support strong conclusions. The dataset contains only 20 tasks, all within a single subdomain of physics, and the evaluation is demonstrated using only two general-purpose LLMs. Several reviewers felt that this scale is insufficient for a benchmark paper at ICLR, and that the work currently reads closer to a technical report or early benchmark release than a fully validated evaluation framework.

After carefully reading the paper, the reviews, and the author rebuttal, I largely agree with the concerns raised by reviewer wzC2. In particular, I agree that the limited task scale, narrow domain coverage, and minimal analysis of rubric validation and inter-judge reliability significantly weaken the contribution. While the benchmark design is reasonable, key aspects such as judge variance, reproducibility of scores, and quality assurance of the rubrics are not evaluated in sufficient depth to support the benchmark’s intended claims.

Other reviewers raised related issues, including limited baseline diversity, unclear handling of multiple valid physical solutions, and the labor-intensive nature of creating gold models and rubrics, which raises questions about scalability. Although the authors acknowledged many of these limitations and outlined future plans, these points remain largely prospective rather than demonstrated. No reviewer followed up after the rebuttal to indicate that these concerns had been alleviated or that they would revise their score.

**Reviewer Scores:**

The reviewer scores were 6, 2, and 6. Reviewers who were more positive emphasized the importance of the problem and the careful benchmark design, while more critical reviewers focused on the limited scale, narrow domain coverage, and insufficient validation of evaluation reliability. As no reviewer followed up, no reviewer explicitly indicated an intention to change their score following the rebuttal or discussion.

---

### Decision · Program_Chairs · 2026-01-26

Reject